# A Scoping Review on Outcomes and Outcome Measurement Instruments in Rehabilitative Interventions for Patients with Haematological Malignancies Treated with Allogeneic Stem Cell Transplantation

**Anastasios I. Manettas [1,2], Panagiotis Tsaklis [2,3,*], Dario Kohlbrenner [4,5] and Lidwine B. Mokkink [6]**

[1] Department of Physiotherapy and Occupational Therapy, University Hospital Zurich, 8091 Zurich, Switzerland; anastasios.manettas@usz.ch

[2] Biomechanics and Ergonomics, ErgoMech Lab, Department of Physical Education and Sport Science, University of Thessaly, 42100 Trikala, Greece

[3] Department of Molecular Medicine and Surgery, Growth and Metabolism, Karolinska Institute, 17176 Stockholm, Sweden

[4] Faculty of Medicine, University of Zurich, 8032 Zurich, Switzerland; dario.kohlbrenner@usz.ch

[5] Department of Pulmonology, University Hospital Zurich, 8091 Zurich, Switzerland

[6] Department of Epidemiology and Data Science, Amsterdam Public Health Research Institute, Amsterdam UMC, Vrije Universiteit Amsterdam, 1007 MB Amsterdam, The Netherlands; w.mokkink@amsterdamumc.nl

**\*** Correspondence: tsaklis@uth.gr

**Abstract:** Rationale: Allogeneic hematopoietic stem cell transplantation (HSCT) is associated with increased treatment-related mortality, loss of physical vitality, and impaired quality of life. Future research will investigate the effects of multidisciplinary rehabilitative interventions in alleviating these problems. Nevertheless, published studies in this field show considerable heterogeneity in selected outcomes and the outcome measurement instruments used. The purpose of this scoping review is to provide an overview of the outcomes and outcome measurement instruments used in studies examining the effects of rehabilitative interventions for patients treated with allogeneic HSCT. Methods: We conducted a scoping review that included randomized controlled trials, pilot studies, and feasibility studies published up to 28 February 2022. Results: We included $n= 39$ studies, in which $n = 84$ different outcomes were used 227 times and $n = 125$ different instruments were used for the measurements. Conclusions: Research in the field of rehabilitation for patients with haematological malignancies treated with allogeneic HSCT is hampered by the excess outcomes used, the inconsistent outcome terminology, and the inconsistent use of measurement instruments in terms of setting and timing. Researchers in this field should reach a consensus with regard to the use of a common terminology for the outcomes of interest and a homogeneity when selecting measurement instruments and measurement timing methods.

**Keywords:** allogeneic stem cell transplantation; outcomes; outcome measurement instruments; rehabilitation; haematological malignancies

## 1. Introduction

### 1.1. Rationale

Allogeneic hematopoietic stem cell transplantation (HSCT) improves the survival rate of patients with haematological malignancies and offers the best chance for cure in a wide range of patients [1,2]. Graft versus Host disease (GvHD) is the most recognized post-allogeneic HSCT complication [3]. Immunosuppressive therapy (IST) is used to treat or prevent both GvHD and further organ damage once GvHD occurs. GvHD and IST are

the two factors most commonly associated with impaired quality of life in these patients [4], distinguishing these patients from those undergoing autologous HSCT. In addition to impaired quality of life, patients treated with allogeneic HSCT for haematological malignancies may have increased treatment-related mortality and loss of physical vitality [5].

Rehabilitation is a complex problem-solving process that is delivered by multidisciplinary teams in inpatient or outpatient settings that aims to improve the patient's quality of life and degree of social integration [6]. Rehabilitative interventions for patients undergoing allogeneic HSCT can improve physical vitality and quality of life as well as decrease mortality [7]. Moreover, early rehabilitation reduces the duration of hospitalization for allogeneic HSCT [8].

Rehabilitative interventions for allogeneic HSCT patients can be challenging with regard to the feasibility of their many phases of treatment. Prior to transplantation, problems related to blood count may not allow the patient to participate in certain rehabilitative interventions. During hospitalization, symptom burden, infections, blood count limitations, or severe fatigue may further prevent the use rehabilitative interventions. Post-hospitalization, GvHD symptoms, blood count fluctuations, or even psychosocial factors may affect the feasibility of rehabilitative interventions. Researchers have long been aware of the importance of the feasibility of rehabilitative interventions among allogeneic HSCT patients [9] and they argue that feasibility and safety should be assessed prior to the development of rehabilitative programs [10].

Future research in this field will investigate the effects of multidisciplinary rehabilitative interventions in a variety of settings. Research in this field has already shown considerable heterogeneity in selected outcomes and in the outcome measurement instruments used [11,12]. Synthesizing, comparing, and interpreting the results from different studies can be challenging when they refer to different outcomes and are measured by different instruments.

*1.2. Objectives*

The purpose of this scoping review is to provide an overview of the outcomes and outcome measurement instruments used in studies examining the effects of rehabilitative interventions for patients treated with allogeneic HSCT, thus enabling a better understanding of the sources of heterogeneity.

**2. Methods**

This scoping review was conducted according to the Preferred Reporting Items for Systematic Reviews and Meta-analyses Extension for Scoping Reviews statement PRISMA ScR (www.prisma-statement.org, (accessed on 28 February 2022)).

*2.1. Data Sources and Study Selection*

The search strategy was developed in collaboration with a librarian to retrieve articles of interest from the MEDLINE, EMBASE, and Cochrane databases in February 2022. Searches were performed with the following terms: (1. xp Hematopoietic Stem Cell Transplantation/ or (transplant* adj5 ("stem cell*" or "hematopoietic cell*" or "haematopoietic cell*")).ti,ab. 2. exp Rehabilitation/ or exp Physical Therapy Modalities/ or exp Exercise/ or exp Exercise Therapy/ or (rehabilitation* or rehabilitative or exercise* or physiotherap* or readaption* or readaptation* or readjustment* or kinesiotherap* or kinesitherap* or training* or (physical adj3 (therap* or treatment*))).ti,ab. 3. (1 and 2) 4. 3 not (animals not humans).sh.). To be included, publications had to be randomized controlled trials, pilot studies, or feasibility studies; published in English or in German; and had to investigate the effects of a rehabilitative intervention shortly before, during, or after allogeneic stem cell transplantation in adult patients with haematological malignancies. After removing duplicates, the titles and abstracts were screened by two

reviewers (AM and DK) against the agreed upon inclusion and exclusion criteria. Studies with no obvious relevance to the research questions were removed. Final inclusion was performed after retrieving and screening the full texts, while disagreements between reviewers were resolved by consensus.

*2.2. Data Extraction, Data Synthesis and Analysis*

The following data were extracted from each article by the lead author: population, intervention, setting, year of publication, country where the research was conducted, outcomes used, outcome measurement instruments used, and timing of measurements. The outcomes and outcome instruments were extracted and classified based on the exact way the authors used them, regardless of the conformity of their terminology with the literature. For example, "aerobic capacity", "peak aerobic capacity", and "functional aerobic capacity" were considered and classified as three different outcomes. The extracted outcomes were not classified as primary or secondary, as this information could not be consistently retrieved from the studies. Furthermore, we classified them according to their measurement core area (Life Impact or Pathophysiological Manifestation) based on the conceptual framework of Boers et al. [13]. According to this framework, outcomes, including the symptoms, signs, events, and biomarkers, that describe how health conditions manifest themselves by abnormal physiology are classified as "Pathophysiological Manifestation" outcomes. Outcomes describing how patients feel, function, or survive are classified as "Life Impact" outcomes. Boers et al. [13] label adverse events separately in their framework in recognition of the prominent role of feasibility in outcome measurements. In this scoping review, we used a third core area to classify all of the feasibility concepts separately. Based on the descriptions of El Kotob et al. [14] and Thabane et al. [15], outcomes describing the feasibility with regard to the safety, processes, resources, and management of a study were classified as "Feasibility" outcomes. Furthermore, the timing of outcome measurements was extracted to show the time-point of the measurements in relation to the day of transplantation and the number of measurements in hierarchical order. We classified the timing of the measurements according to hospital or non-hospital settings.

## 3. Results

Our search yielded a total of $n$ = 1781 publications after duplicates were removed (Figure 1). Of these, we assessed $n$ = 195 for eligibility based on the full text. A total of $n$ = 39 studies of the following types met the inclusion criteria [16–54]: $n$ = 24 randomized clinical trials, $n$ = 9 pilot studies, and $n$ = 6 feasibility studies (Table 1). In these 39 studies, $n$ = 84 different outcomes were used 227 times and were measured using $n$ = 125 different instruments (Table 2).

**Table 1.** Study characteristics.

| Study Characteristics | |
|---|---|
| Included Studies | $N$ = 39 (YY%) |
| Study Design | |
| RCTs | 24 (62%) |
| Feasibility Studies | 9 (23%) |
| Pilot Studies | 6 (15%) |
| Population | |
| Allogeneic stem cell transplantation | 14 (36%) |
| Allogeneic and autologous stem cell transplantation | 25 (64%) |
| Setting | |
| Hospital | 22 (56%) |

| | |
|---|---|
| Outpatient post HSCT | 11 (28%) |
| Outpatient pre HSCT | 3 (8%) |
| Throughout | 2 (5%) |
| Inpatient post HSCT | 1 (3%) |
| Intervention | |
| Psychological Interventions | 5 (13%) |
| Exercise Training | 17 (44%) |
| Respiratory Training | 3 (8%) |
| Physical Modalities | 2 (5%) |
| Relaxation Techniques | 2 (5%) |
| Other | 10 (25%) |
| Language | |
| English | 39 (100%) |
| Country | |
| USA | 13 (34%) |
| Germany | 9 (23%) |
| Canada | 4 (10%) |
| Brazil | 4 (10%) |
| Switzerland | 2 (5%) |
| Other | 7 (18%) |

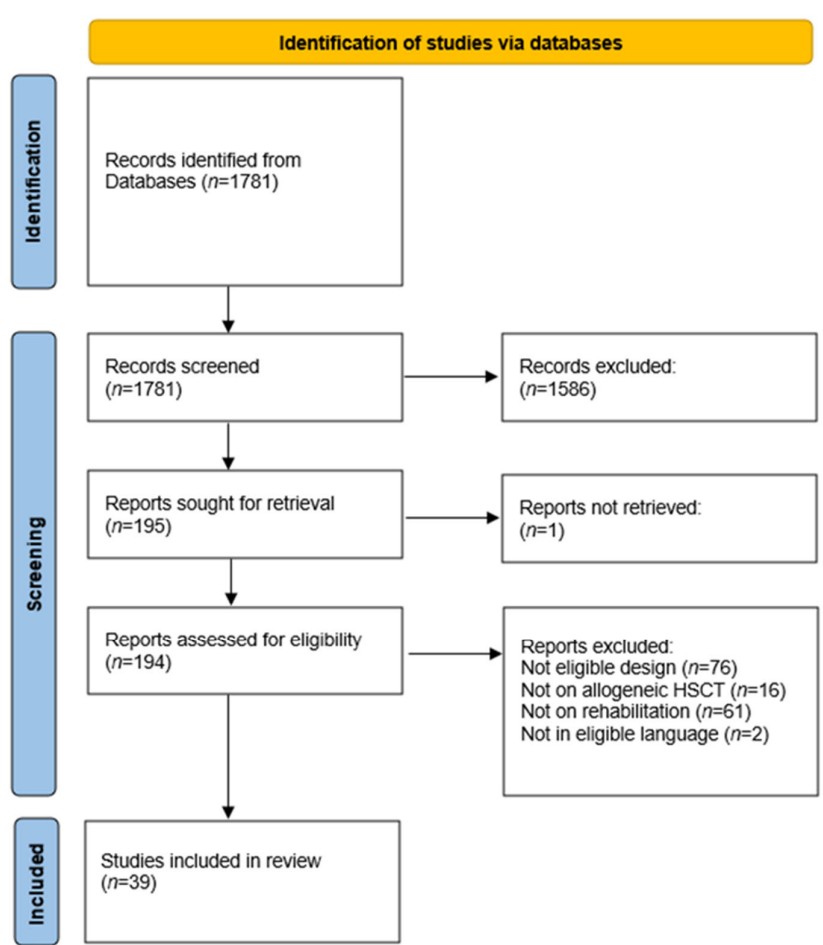

**Figure 1.** Study selection flowchart.

<p align="center">**Table 2.** Outcomes and outcome measurement instruments.</p>

**CORE AREA «FEASIBILITY»**

Allogeneic

| Outcomes | Instruments | Design | Phase | Intervention | Reference |
|---|---|---|---|---|---|
| Feasibility [16,17] *N* = 2 | Adherence [16] Adverse Events [16] Program completion from 50% of the patients [17] Recruitment [16] | Feasibility Pilot Study [17] | Hospital [17] | Exercise [16,17] | Santa mina et al., 2020 Schuler et al., 2016 |

HSCT

| Outcomes | Instruments | Design | Phase | Intervention | Reference |
|---|---|---|---|---|---|
| Feasibility [18–25,27] *N* = 7 | Adherence [18–21,25,27] | Feasibility [18–20] Pilot Study [21,25] | Hospital [18–20] Outpatient post [21] Outpatient pre [25] | Electric Muscle Stimulation [18] Healing touch [19] Inspiratory muscle training [20] Home based aerobic exercise [6] Exercise Training and Nutritional Support [25] | Bewarder et al., 2019 Lu et al., 2016 De almeida et al., 2020 Wilson et al., 2005 Rupnik et al., 2020 |
| | Attrition [20,21,25,27] | Feasibility [20] Pilot Study [21,25] | Hospital [20] Outpatient post [21] Outpatient pre [25] | Inspiratory muscle training [20] Home based aerobic exercise [21] Exercise Training and Nutritional Support [25] | De almeida et al., 2020 Wilson et al., 2005 Rupnik et al., 2020 |
| | Retention [19,22] | Feasibility [19,22] | Hospital [19] Outpatient post [22] | Healing touch [19] Yoga [22] | Lu et al., 2016 Baydoun et al., 2020 |
| | Acceptability [21,25] | Pilot Study [21,25] | Outpatient post [21] Outpatient pre [25] | Home based aerobic exercise [21] Exercise Training and Nutritional Support [25] | Wilson et al., 2005 Rupnik et al., 2020 |
| | Accrual Acceptance [22] | Feasibility [22] | Outpatient post [22] | Yoga [22] | Baydoun et al., 2020 |
| | Adverse Events [22] | Feasibility [22] | Outpatient post [22] | Yoga [22] | Baydoun et al., 2020 |
| | N/A [23] | RCT [23] | Hospital [23] | Exergaming [23] | Schumacher et al., 2018 |
| | Protocol Adherence [22] | Feasibility [22] | Outpatient post [22] | Yoga [22] | Baydoun et al., 2020 |
| | Rate of participant enrolment [24] | Pilot Study [24] | Outpatient post [24] | Positive Psychology Intervention [24] | Amonoo et al., 2020 |
| | Rate of session completion [24] | Pilot Study [24] | Outpatient post [24] | Positive Psychology Intervention [24] | Amonoo et al., 2020 |

| Outcomes | Instruments | Design | Phase | Intervention | Reference |
|---|---|---|---|---|---|
| | Recruitment [19,27] | Feasibility [19] | Hospital [19] | Healing touch [19] | Lu et al., 2016 |
| | Recruitment Rate [20] | Feasibility [20] | Hospital [20] | Inspiratory muscle training [20] | De almeida et al., 2020 |
| Safety [18,20,21] *N* = 4 | Adverse Events [18,20,21] The WHO bleeding Scale [18] | Feasibility [18,20] Pilot Study [21] | Hospital [18,20] Outpatient post [21] | Electric Muscle Stimulation [18] Inspiratory muscle training [20] Home based aerobic exercise [21] | Bewarder et al., 2019 De almeida et al., 2020 Wilson et al., 2005 |
| Acceptability [24] *N* = 1 | Rating of ease and utility [24] | Pilot Study [24] | Outpatient post [24] | Positive Psychology Intervention [24] | Amonoo et al., 2020 |
| Adherence [26] *N* = 1 | Exercise Sessions completed as proportion of the prescribed exercises [26] | RCT [26] | Throughout [26] | Exercise Training [26] | Peters et al., 2018 |
| Attrition [26] *N* = 1 | N/A [26] | RCT [26] | Throughout [26] | Exercise Training [26] | Peters et al., 2018 |
| Compliance [26] *N* = 1 | Exercise Sessions completed [26] | RCT [26] | Throughout [26] | Exercise Training [26] | Peters et al., 2018 |
| Progression after initial prescription [26] *N* = 1 | Added sets, repetitions or exercises [26] | RCT [26] | Throughout [26] | Exercise Training [26] | Peters et al., 2018 |
| Adverse Events [27] | | | | | Fioritto et al., 2021 |

CORE AREA LIFE IMPACT

Allogeneic

| Outcomes | Instruments | Design | Phase | Intervention | Reference |
|---|---|---|---|---|---|
| Fatigue [16,17,28–33] *N* = 8 | Brief Fatigue Inventory [28,29] | Pilot Study [28] RCT [29] | Outpatient post [28,29] | Individualized Exercise Program [28] Supervised exercise program [29] | Carlson et al., 2006 Shleton et al., 2008 |
| | FACT-F [16,28] | Feasibility [16] Pilot Study [28] | Hospital [16] Outpatient post [28] | Exercise [16] Individualized Exercise Program [28] | Santa mina et al., 2020 Carlson et al., 2006 |
| | Multidimensional Fatigue Inventory [16,30] | Feasibility [16] RCT [30] | Hospital [16,30] | Exercise [16] Whole Body Vibration Training [30] | Santa mina et al., 2020 Pahl et al., 2020 |
| | EORTC QLQ FA-13 [17] | Pilot Study [17] | Hospital [17] | Exercise [17] | Schuler et al., 2016 |
| | FACT-An Anemia Scale [31] | RCT [31] | Hospital [31] | Multimodal Intervention [31] | Jarden et al., 2009 |
| | Fatigue Impact Scale (FIS) [32] | RCT [32] | Inpatient post [32] | Inspiratory muscle training [32] | Bargi et al., 2015 |
| | Piper Fatigue Scale [33] | RCT [33] | Hospital [33] | Relaxation Breathing Exercise [33] | Kim et al., 2005 |

| Depression [16,17,28,32,33] N = 5 | Hospital Anxiety And Depression Scale [17] | Pilot Study [17] | Hospital [17] | Exercise [17] | Schuler et al., 2016 |
|---|---|---|---|---|---|
| | Montgomery-Âsberg Depression Rating Scale (MADRS) [32] | RCT [32] | Inpatient post [32] | Inspiratory muscle training [32] | Bargi et al., 2015 |
| | Structured Clinical Interview [28] | Pilot Study [28] | Outpatient post [28] | Individualized Exercise Program [28] | Carlson et al., 2006 |
| | The Beck Depression Inventory [33] | RCT [33] | Hospital [33] | Relaxation Breathing Exercise [33] | Kim et al., 2005 |
| | The Center for Epidemiological Studies Depression Scale (CES-D) [28] | Pilot Study [28] | Outpatient post [28] | Individualized Exercise Program [28] | Carlson et al., 2006 |
| | The Patient Health Questionnaire 9 (PHQ-9) [16] | Feasibility [16] | Hospital [16] | Exercise [16] | Santa mina et al., 2020 |
| Quality of Life [17,30,32,34,35] N = 5 | EORTC QLQ-C30 [17,30,32,34] | Pilot Study [17] RCT [30,32,34] | Hospital [17,30,34] Inpatient post [32] | Exercise [17] Inspiratory muscle training [32] Exercise Training [34] Whole Body Vibration Training [30] | Schuler et al., 2016 Bargi et al., 2015 Baumann et al., 2011 Pahl et al., 2020 |
| | FACT-BMT [35] | Feasibility [35] | Outpatient post [35] | Telehealth Psychoeducational support [35] | Lounsbery et al., 2010 |
| Health related Quality of Life [16,31,36,37] N = 4 | EORTC QLQ-C30 [16,31,36,37] | Feasibility [16] RCT [31,36,37] | Hospital [16,31,36] Throughout [37] | Exercise [16] Multimodal Intervention [31] Whole Body Vibration Training [36] Self-administered exercise [37] | Santa mina et al., 2020 Kaeding et al., 2018 Wiskemann et al., 2011 |
| Anxiety [16,33,38] N = 3 | Generalized Anxiety Disorder GAD7 [16] | Feasibility [16] | Hospital [16] | Exercise [16] | Santa mina et al., 2020 |
| | The State-Trait Anxiety Inventory [33] | RCT [33] | Hospital [33] | Relaxation Breathing Exercise [33] | Kim et al., 2005 |
| | Visual Analog Scale [38] | RCT [38] | Hospital [38] | Music Therapy [38] | Doro et al., 2017 |
| Mood [28,38] N = 2 | Profile of Mood States (POMS) [28] | Pilot Study [28] | Outpatient post [28] | Individualized Exercise Program [28] | Carlson et al., 2006 |
| | Visual Analog Scale [38] | RCT [38] | Hospital [38] | Music Therapy [38] | Doro et al., 2017 |

| Distress [37] N = 1 | The National Comprehensive Cancer Network Distress Thermometer [37] | RCT [37] | Throughout [37] | Self-administered exercise [37] | Wiskemann et al., 2011 |
|---|---|---|---|---|---|
| Perception of Personal Benefits [35] N = 1 | Post Traumatic growth inventory [35] | Feasibility [35] | Outpatient post [35] | Telehealth Psychoeducational support [35] | Lounsbery et al., 2010 |
| Physical Activity [30] N = 1 | Freiburg Questionnaire on physical activity [30] | RCT [30] | Hospital [30] | Whole Body Vibration Training [30] | Pahl et al., 2020 |
| Physical Well Being [37] N = 1 | Hospital Anxiety And Depression Scale [37] | RCT [37] | Throughout [37] | Self-administered exercise [37] | Wiskemann et al., 2011 |
| Psychological Well Being [31] N = 1 | Hospital Anxiety And Depression Scale [31] | RCT [31] | Hospital [31] | Multimodal Intervention [31] | Jarden et al., 2009 |
| Self-efficacy for exercise [16] N = 1 | Exercise Self Efficacy Scale [16] | Feasibility [16] | Hospital [16] | Exercise [16] | Santa mina et al., 2020 |
| Spirituality and Meaning Making [35] N = 1 | FACT-SP [35] | Feasibility [35] | Outpatient post [35] | Telehealth Psychoeducational support [35] | Lounsbery et al., 2010 |
| Subjective Distress [35] N = 1 | Impact of Event Scale Revised [35] | Feasibility [35] | Outpatient post [35] | Telehealth Psychoeducational support [35] | Lounsbery et al., 2010 |

| HSCT | | | | | |
|---|---|---|---|---|---|
| Outcomes | Instruments | Design | Phase | Intervention | Reference |
| Fatigue [21,22,39–43] N = 7 | Brief Fatigue Inventory [39] | RCT [39] | Hospital [39] | Relaxation Techniques [39] | Jafari et al., 2018 |
| | Chalder Fatigue Scale [40] | RCT [40] | Hospital [40] | Strength Training [40] | Hacker et al., 2017 |
| | EORTC QLQ-C30 [40] | RCT [40] | Hospital [40] | Strength Training [40] | Hacker et al., 2017 |
| | FACT-An Anaemia Scale [41] | RCT [41] | Outpatient post [41] | Outpatient physical exercise [41] | Knols et al., 2011 |
| | Fatigue Symptom Inventory [21] | Pilot Study [21] | Outpatient post [21] | Home based aerobic exercise [21] | Wilson et al., 2005 |
| | Multidimensional Fatigue Inventory [22] | Feasibility [22] | Outpatient post [22] | Yoga [22] | Baydoun et al., 2020 |
| | SF36 [42] | RCT [42] | Outpatient post [42] | Exercise Relaxation Information [42] | Bird et al., 2010 |
| | The Fatigue Questionnaire (FQ) [43] | Feasibility [43] | Outpatient post [43] | Mindfulness based Intervention [43] | Grossman et al., 2015 |

| Depression [19,23,43–46] N = 6 | Hospital Anxiety And Depression Scale [23,44–46] | RCT [23,44–46] | Hospital [23,44–46] | Exergaming [23] | Schumacher et al., 2018 |
|---|---|---|---|---|---|
| | | | | Media Art [44] | Mc Cabe et al., 2013 |
| | | | | Palliative Care [45] | El Jawahri et al., 2017 |
| | | | | Problem Solving Training [46] | Balck et al., 2019 |
| | The Centre for Epidemiological Studies Depression Scale (CES-D) [19,43] | Feasibility [19,43] | Hospital [19] | Healing touch [19] | Lu e al 2016 |
| | | | Outpatient post [43] | Mindfulness based Intervention [43] | Grossman et al., 2015 |
| Anxiety [23,43–46] N = 5 | Hospital Anxiety And Depression Scale [23,44,46] | RCT [23,44–46] | Hospital [23,44–46] | Exergaming [23] | Schumacher et al., 2018 |
| | | | | Media Art [44] | Mc Cabe et al., 2013 |
| | | | | Palliative Care [45] | El Jawahri et al., 2017 |
| | | | | Problem Solving Training [46] | Balck et al., 2019 |
| | The Spielberger Trait Anxiety Scale (STAI) [43] | Feasibility [43] | Outpatient post [43] | Mindfulness based Intervention [43] | Grossman et al., 2015 |
| Quality of Life [19,34,40,42,45] N = 5 | EORTC QLQ-C30 [34,40] | RCT [34,40] | Hospital [34,40] | Exercise Therapy [34] | Baumann et al., 2010 |
| | | | | Strength Training [40] | Hacker et al., 2017 |
| | FACT-BMT [19,45] | Feasibility [19] | Hospital [19] | Healing touch [19] | Lu e al 2016 |
| | | RCT [45] | Hospital [45] | Palliative Care [45] | El Jawahri et al., 2017 |
| | The Graham and Longman QoL Questionnaire [42] | RCT [42] | Outpatient post [42] | Exercise Relaxation Information [42] | Bird et al., 2010 |
| Health related Quality of Life [21,23,25,27,41,43,47] N = 7 | EORTC QLQ-C30 [25,27,41] | RCT [41] Pilot [25] Feasibility [27] | Outpatient post [41] Outpatient pre [25] | Outpatient physical exercise [41] Exercise Training and Nutritional Support [25] Individualized Exercise Training [27] | Knols et al., 2011 Rupnik et al., 2020 Fioritto et al., 2021 |
| | FACT [43] | Feasibility [43] | Outpatient post [43] | Mindfulness based Intervention [43] | Grossman et al., 2015 |

| | | | | | |
|---|---|---|---|---|---|
| | FACT-BMT [23,47] | RCT [23,47] | Hospital [23,47] | Exergaming [23] Multidirectional Walking [47] | Schumacher et al., 2018 Potiaumpai et al., 2020 |
| | Profile of Health-Related Quality of Life in Chronic Disorders [43] | Feasibility [43] | Outpatient post [43] | Mindfulness based Intervention [43] | Grossman et al., 2015 |
| | SF36 [21,23] | Pilot Study [21] | Outpatient post [21] | Home based aerobic exercise [21] | Wilson et al., 2005 |
| | | RCT [23] | Hospital [23] | Exergaming [23] | Schumacher et al., 2018 |
| Physical Activity [40,41] 3 | Accelerometry [40] | RCT [40] | Hospital [40] | Strength Training [40] | Hacker et al., 2017 |
| | International Physical Activity Questionnaire [41] | RCT [41] | Outpatient post [41] | Outpatient physical exercise [41] | Knols et al., 2011 |
| | The Godin leisure time exercise questionnaire [40] | RCT [40] | Hospital [40] | Strength Training [40] | Hacker et al., 2017 |
| Distress [19,44] *N* = 2 | Profile of Mood States (POMS) [19] | Feasibility [19] | Hospital [19] | Healing touch [19] | Lu et al., 2016 |
| | The Distress Thermometer [44] | RCT [44] | Hospital [44] | Media Art [44] | Mc Cabe et al., 2013 |
| Post-traumatic Stress Disorder Symptoms [45,48] *N* = 2 | Clinician administered PTSD Scale for DSM IV [48] | RCT [48] | Outpatient post [48] | Cognitive Behavioural Therapy [48] | DuHamel et al., 2010 |
| | PCL-C [45,48] | RCT [45,48] | Hospital [45] | Palliative Care [45] | El Jawahri et al., 2017 |
| | | | Outpatient post [33] | Cognitive Behavioural Therapy [48] | DuHamel et al., 2010 |
| Psychological distress [46,48] *N* = 2 | Brief Symptom Inventory [48] | RCT [48] | Outpatient post [48] | Cognitive Behavioural Therapy [48] | DuHamel et al., 2010 |
| | SCL-K-9 [46] | RCT [46] | Hospital [46] | Problem Solving Training [46] | Balck et al., 2019 |
| Bodily Pain [42] | SF36 [42] | RCT [42] | Outpatient post [42] | Exercise Relaxation Information [42] | Bird et al., 2010 |
| Coping [46] *N* = 1 | The Brief COPE [46] | RCT [46] | Hospital [46] | Problem Solving Training [46] | Balck et al., 2019 |
| Extend of the pain [46] *N* = 1 | The Questions of Pain [46] | RCT [46] | Hospital [46] | Problem Solving Training [46] | Balck et al., 2019 |

| | | | | | |
|---|---|---|---|---|---|
| General Distress [46] *N* = 1 | The National Comprehensive Cancer Network Distress Thermometer [46] | RCT [46] | Hospital [46] | Problem Solving Training [46] | Balck et al., 2019 |
| General distress and depressive symptoms [49] | Brief Symptom Inventory [49] | RCT [49] | Outpatient post [49] | Telephone administered cognitive behavioural therapy [49] | Applebaum et al., 2012 |
| General Mental Health [42] *N* = 1 | SF36 [42] | RCT [42] | Outpatient post [42] | Exercise Relaxation Information [42] | Bird et al., 2010 |
| Illness Related PTSD Symptoms [49] | PCL-C [49] | RCT [49] | Outpatient post [49] | Telephone administered cognitive behavioural therapy [49] | Applebaum et al., 2012 |
| Mental Well Being [50] | Cancer and Treatment Distress [50] | RCT [50] | Outpatient pre [50] | Exercise and Stress Management [50] | Jacobsen et al., 2014 |
| | Pittsburgh Sleep Quality Index [35] | RCT [35] | Outpatient pre [35] | Exercise and Stress Management [35] | Jacobsen et al., 2014 |
| Physical Fitness [8] | Human Activity Profile [8] | RCT [8] | Hospital [8] | Exergaming [8] | Schumacher et al., 2018 |
| Physical Functioning [27] | SF36 [27] | RCT [27] | Outpatient post [27] | Exercise Relaxation Information [27] | Bird et al., 2010 |
| Physical Well Being [35] | SF36 [35] | RCT [35] | Outpatient pre [35] | Exercise and Stress Management [35] | Jacobsen et al., 2014 |
| Problem Solving Ability [46] *N* = 1 | The Social Problem-Solving Inventory-Revised (SPSI-R) [46] | RCT [46] | Hospital [46] | Problem Solving Training [46] | Balck et al., 2019 |
| Psychological health [42] | General Health Questionnaire [42] | RCT [42] | Outpatient post [42] | Exercise Relaxation Information [42] | Bird et al., 2010 |
| Psychological Performance [18] | EORTC QLQ-C30 [18] | Feasibility [18] | Hospital [18] | Electric Muscle Stimulation [18] | Bewarder et al., 2019 |
| | Multidimensional Fatigue Inventory [18] | Feasibility [18] | Hospital [18] | Electric Muscle Stimulation [18] | Bewarder et al., 2019 |
| Quantified Walking Activity [41] | Accelerometry [41] | RCT [41] | Outpatient post [41] | Outpatient physical exercise [41] | Knols et al., 2011 |
| Role Limitation [42] | SF36 [42] | RCT [42] | Outpatient post [42] | Exercise Relaxation Information [42] | Bird et al., 2010 |

| Self-Reported Physical Function [41] | EORTC QLQ-C30 [41] | RCT [41] | Outpatient post [41] | Outpatient physical exercise [41] | Knols et al., 2011 |
|---|---|---|---|---|---|
| Social Functioning [42] | SF36 [42] | RCT [42] | Outpatient post [42] | Exercise Relaxation Information [42] | Bird et al., 2010 |
| Social support [46] *N* = 1 | N/A | RCT [46] | Hospital [46] | Problem Solving Training [46] | Balck et al., 2019 |
| Symptom Burden [45] *N* = 1 | Edmonton Symptom Assessment Scale [45] | RCT [45] | Hospital [45] | Palliative Care [45] | El Jawahri et al., 2017 |
| Therapeutic alliance [49] | Working Alliance Inventory Short Form [49] | RCT [49] | Outpatient post [49] | Telephone-administered cognitive behavioural therapy [49] | Applebaum et al., 2012 |
| Vitality [42] | SF36 [42] | RCT [42] | Outpatient post [42] | Exercise Relaxation Information [42] | Bird et al., 2010 |

CORE AREA PATHOPHYSIOLOGY

Allogeneic

| Outcomes | Instruments | Design | Phase | Intervention | Reference |
|---|---|---|---|---|---|
| Endurance [17,51] *N* = 2 | Bicycle ergometer [17] | Pilot Study [17] | Hospital [17] | Exercise [17] | Schuler et al., 2016 |
| | The 6-Minute Walk Test [17] | Pilot Study [17] | Hospital [17] | Exercise [17] | Schuler et al., 2016 |
| | The WHO Scheme [51] | RCT [51] | Hospital [51] | Exercise Training [51] | Baumann et al., 2011 |
| Functional Performance [30,31] *N* = 2 | 2 min stair climb test [31] | RCT [31] | Hospital [31] | Multimodal Intervention [31] | Jarden et al., 2009 |
| | Chairing test on force plate [30] | RCT [30] | Hospital [30] | Whole Body Vibration Training [30] | Pahl et al., 2020 |
| | Maximum Counter movement jump [30] | RCT [30] | Hospital [30] | Whole Body Vibration Training [30] | Pahl et al., 2020 |
| Handgrip Strength [16,32] *N* = 2 | Hand Grip Dynamometer [16,32] | Feasibility [16] RCT [32] | Hospital [16] Inpatient post [32] | Exercise [16] Inspiratory muscle training [32] | Santa mina et al., 2020 Bargi et al., 2015 |
| Physical Performance [28,29] *N* = 2 | 50-foot walk test [29] | RCT [29] | Outpatient post [29] | Supervised exercise program [29] | Shleton et al., 2008 |
| | Blood Lactate Concentrate [28] | Pilot Study [28] | Outpatient post [28] | Individualized Exercise Program [28] | Carlson et al., 2006 |
| | Cardiac Output [28] | Pilot Study [28] | Outpatient post [28] | Individualized Exercise Program [28] | Carlson et al., 2006 |

| | Forward reach [29] | RCT [29] | Outpatient post [29] | Supervised exercise program [29] | Shleton et al., 2008 |
|---|---|---|---|---|---|
| | Oxygen Uptake (VO2) [28] | Pilot Study [28] | Outpatient post [28] | Individualized Exercise Program [28] | Carlson et al., 2006 |
| | Power Output [28] | Pilot Study [28] | Outpatient post [28] | Individualized Exercise Program [28] | Carlson et al., 2006 |
| | Rating of Perceived Exertion (RPE) [28] | Pilot Study [28] | Outpatient post [28] | Individualized Exercise Program [28] | Carlson et al., 2006 |
| | Respiratory Exchange Ratio [28] | Pilot Study [28] | Outpatient post [28] | Individualized Exercise Program [28] | Carlson et al., 2006 |
| | Stroke Volume [28] | Pilot Study [28] | Outpatient post [28] | Individualized Exercise Program [28] | Carlson et al., 2006 |
| | The 6-Minute Walk Test [29] | RCT [29] | Outpatient post [29] | Supervised exercise program [29] | Shleton et al., 2008 |
| | Timed repeated sit to stand [29] | RCT [29] | Outpatient post [29] | Supervised exercise program [29] | Shleton et al., 2008 |
| | Uniped stance time [29] | RCT [29] | Outpatient post [29] | Supervised exercise program [29] | Shleton et al., 2008 |
| | Ventilatory Threshold [28] | Pilot Study [28] | Outpatient post [28] | Individualized Exercise Program [28] | Carlson et al., 2006 |
| Pulmonary Function [32,51] *N* = 2 | Spirometry [32,51] | RCT [32,51] | Hospital [51] Inpatient post [32] | Exercise Training [51] Inspiratory muscle training [32] | Baumann et al., 2011 Bargi et al., 2015 |
| Aerobic endurance performance capacity [36] | The 6-Minute Walk Test [36] | RCT [36] | Hospital [36] | Whole Body Vibration Training [36] | Kaeding et al., 2018 |
| Body Composition [30] | Displacement Plethysmography [30] | RCT [30] | Hospital [30] | Whole Body Vibration Training [30] | Pahl et al., 2020 |
| Body Mass Index [16] | Bioimpendance Analysis [16] | Feasibility [16] | Hospital [16] | Exercise [16] | Santa mina et al., 2020 |
| Cardiorespiratory Fitness [30] | Cardiopulmonary Exercise Testing [30] | RCT [30] | Hospital [30] | Whole Body Vibration Training [30] | Pahl et al., 2020 |
| Functional Aerobic Capacity [16] | 30 Second Sit to Stand Test [16] | Feasibility [16] | Hospital [16] | Exercise [16] | Santa mina et al., 2020 |
| | The 6-Minute Walk Test [16] | Feasibility [16] | Hospital [16] | Exercise [16] | Santa mina et al., 2020 |
| Functional Status [52] | Karnofsky Performance Status Scale [52] | RCT [52] | Hospital [52] | Walking Regimen [52] | DeFor et al., 2007 |

| Isokinetic Leg Performance [36] | Biodex System [36] | RCT [36] | Hospital [36] | Whole Body Vibration Training [36] | Kaeding et al., 2018 |
|---|---|---|---|---|---|
| Knee Extension Strength [17] | External resistor [17] | Pilot Study [17] | Hospital [17] | Exercise [17] | Schuler et al., 2016 |
| Leucocyte count [33] | Total and differential Counts of white blood cells [33] | RCT [33] | Hospital [33] | Relaxation Breathing Exercise [33] | Kim et al., 2005 |
| Maximal Exercise Capacity [32] | The Modified Incremental Shuttle Walking Test (MISWT) [32] | RCT [32] | Inpatient post [32] | Inspiratory muscle training [32] | Bargi et al., 2015 |
| Muscle Strength [31] | Isotonic muscular strength [31] | RCT [31] | Hospital [31] | Multimodal Intervention [31] | Jarden et al., 2009 |
| | Maximal isometric voluntary strength [31] | RCT [31] | Hospital [31] | Multimodal Intervention [31] | Jarden et al., 2009 |
| Pain [38] | Visual Analog Scale [38] | RCT [38] | Hospital [38] | Music Therapy [38] | Doro et al., 2017 |
| Peak aerobic capacity [16] | Cardiopulmonary Exercise Testing [16] | Feasibility [16] | Hospital [16] | Exercise [16] | Santa mina et al., 2020 |
| Peak Oxygen Consumption [30] | Cardiopulmonary Exercise Testing [30] | RCT [30] | Hospital [30] | Whole Body Vibration Training [30] | Pahl et al., 2020 |
| Peripheral Muscle Strength [32] | Hand-Held Dynamometer [32] | RCT [32] | Inpatient post [32] | Inspiratory muscle training [32] | Bargi et al., 2015 |
| Physical Capacity [31] | Estimated VO2max cycle ergometer [31] | RCT [31] | Hospital [31] | Multimodal Intervention [31] | Jarden et al., 2009 |
| Respiratory Muscle Strength [32] | Mouthpiece device [32] | RCT [32] | Inpatient post [32] | Inspiratory muscle training [32] | Bargi et al., 2015 |
| Strength [51] *N* = 1 | Isometric Test Digimax [51] | RCT [51] | Hospital [51] | Exercise Training [51] | Baumann et al., 2011 |
| Strength Capacity [30] | Isokinetic Test Knee Extensors [30] | RCT [30] | Hospital [30] | Whole Body Vibration Training [30] | Pahl et al., 2020 |
| Submaximal Exercise Capacity [32] | The 6-Minute Walk Test [32] | RCT [32] | Inpatient post [32] | Inspiratory muscle training [32] | Bargi et al., 2015 |
| Trunk strength [17] | N/A | Pilot Study [17] | Hospital [17] | Exercise [17] | Schuler et al., 2016 |
| Upper Limb Muscle Strength [16] | Hand-Held Dynamometer [16] | Feasibility [16] | Hospital [16] | Exercise [16] | Santa mina et al., 2020 |
| HSCT | | | | | |
| Outcomes | Instruments | Design | Phase | Intervention | Reference |
| Endurance [23,34] *N* = 2 | The 2 Minute Walk Test [23] | RCT [23] | Hospital [23] | Exergaming [23] | Schumacher et al., 2018 |
| | The WHO Scheme [34] | RCT [34] | Hospital [34] | Exercise Therapy [34] | Baumann et al., 2010 |
| | Treadmill [23] | RCT [23] | Hospital [23] | Exergaming [23] | Schumacher et al., 2018 |

| Handgrip Strength [23,41] *N* = 2 | Hand Grip Dynamometer [23,41] | RCT [23,41] | Hospital [23] Outpatient post [41] | Exergaming [23] Outpatient physical exercise [41] | Schumacher et al., 2018 Knols et al., 2011 |
|---|---|---|---|---|---|
| Aerobic Fitness [21] | Ventilatory Threshold [21] | Pilot Study [21] | Outpatient post [21] | Home based aerobic exercise [21] | Wilson et al., 2005 |
| Blood count [34] | *N/A* | RCT [34] | Hospital [34] | Exercise Therapy [34] | Baumann et al., 2010 |
| Body Composition [25,41] | Dual x-ray absorptiometry [41] | RCT [41] | Outpatient post [41] | Outpatient physical exercise [41] | Knols et al., 2011 |
| | Bioimpedance Analysis [25] | Pilot Study [25] | Outpatient pre [25] | Exercise Training and Nutritional Support [25] | Rupnik et al., 2020 |
| Cardiorespiratory Fitness [53] | Cardiopulmonary Exercise Testing [53] | Feasibility [53] | Outpatient pre [53] | Home based interval exercise training [53] | Wood et al., 2016 |
| | The 6-Minute Walk Test [53] | Feasibility [53] | Outpatient pre [53] | Home based interval exercise training [53] | Wood et al., 2016 |
| Exercise Capacity [42] | Shuttle Walk Test (SWT) [42] | RCT [42] | Outpatient post [42] | Exercise Relaxation Information [42] | Bird et al., 2010 |
| Functional Ability [40] | 15 Foot Walk Time [40] | RCT [40] | Hospital [40] | Strength Training [40] | Hacker et al., 2017 |
| | 30 Second Sit to Stand Test [40] | RCT [40] | Hospital [40] | Strength Training [40] | Hacker et al., 2017 |
| | Timed Stair Climb [40] | RCT [40] | Hospital [40] | Strength Training [40] | Hacker et al., 2017 |
| | Timed Up and Go Test [40] | RCT [40] | Hospital [40] | Strength Training [40] | Hacker et al., 2017 |
| Functional Exercise Capacity [41] | The 6-Minute Walk Test [41] | RCT [41] | Outpatient post [41] | Outpatient physical exercise [41] | Knols et al., 2011 |
| Functional Capacity [27] | 6 min step test [27] | Feasibility [27] | Hospital [27] | Individualized Exercise Training [27] | Fioritto et al., 2021 |
| Knee Extension Strength [41] | Hand-Held Dynamometer [41] | RCT [41] | Outpatient post [41] | Outpatient physical exercise [41] | Knols et al., 2011 |
| Muscle Strength [40] [25] | Arm curl test [40] | RCT [40] | Hospital [40] | Strength Training [40] | Hacker et al., 2017 |
| | Hand Grip Dynamometer [25,40] | RCT [40] Pilot Study [25] | Hospital [40] Outpatient pre [25] | Strength Training [40] Exercise Training and Nutritional Support [25] | Hacker et al., 2017 Rupnik et al., 2020 |

| | Rectus femoris cross sectional area [40] | RCT [40] | Hospital [40] | Strength Training [40] | Hacker et al., 2017 |
|---|---|---|---|---|---|
| Nausea [35] | SF36 [50] | RCT [50] | Outpatient pre [50] | Exercise and Stress Management [50] | Jacobsen et al., 2014 |
| Physical Performance [18] [25] | 8 Foot Walk [18] | Feasibility [18] | Hospital [18] | Electric Muscle Stimulation [18] | Bewarder et al., 2019 |
| | Balance Test [18] | Feasibility [18] | Hospital [18] | Electric Muscle Stimulation [18] | Bewarder et al., 2019 |
| | Chair Stands [18] | Feasibility [18] | Hospital [18] | Electric Muscle Stimulation [18] | Bewarder et al., 2019 |
| | The 6-Minute Walk Test [18,25] | Feasibility [18] Pilot Study [25] | Hospital [18] Outpatient pre [25] | Electric Muscle Stimulation [18] Exercise Training and Nutritional Support [25] | Bewarder et al., 2019 Rupnik et al., 2020 |
| | The Short Physical Performance Battery [18] | Feasibility [18] | Hospital [18] | Electric Muscle Stimulation [18] | Bewarder et al., 2019 |
| | 30 Second Sit to Stand Test [25] | Pilot Study [25] | Outpatient pre [25] | Exercise Training and Nutritional Support [25] | Rupnik et al., 2020 |
| Pulmonary Function [34] | Spirometry [34] | RCT [34] | Hospital [34] | Exercise Therapy [34] | Baumann et al., 2010 |
| Respiratory Function [54] | Tidal Volume, minute volume, maximal inspiratory and expiratory pressures [54] | Pilot Study [54] | Hospital [54] | Respiratory Physiotherapy [54] | Bom et al., 2012 |
| Respiratory Muscle Strength [20] | Maximal Expiratory Pressure [20] | Feasibility [20] | Hospital [20] | Inspiratory muscle training [20] | De almeida et al., 2020 |
| | Maximal Inspiratory Pressure [20] | Feasibility [20] | Hospital [20] | Inspiratory muscle training [20] | De almeida et al., 2020 |
| Respiratory Signs [20] | Peripheral Oxygen Saturation [20] | Feasibility [20] | Hospital [20] | Inspiratory muscle training [20] | De almeida et al., 2020 |
| | Respiratory Rate [20] | Feasibility [20] | Hospital [20] | Inspiratory muscle training [20] | De almeida et al., 2020 |
| Respiratory Symptoms [20] | Medical Records [20] | Feasibility [20] | Hospital [20] | Inspiratory muscle training [20] | De almeida et al., 2020 |
| Strength [34] | Isometric Test Digimax [34] | RCT [34] | Hospital [34] | Exercise Therapy [34] | Baumann et al., 2010 |
| Walking Speed [41] | 50-foot walk test [41] | RCT [41] | Outpatient post [41] | Outpatient physical exercise [41] | Knols et al., 2011 |

| Upper Limb Muscle Strength [27] | Handgrip Dynamometer [27] | Feasibility [27] | Hospital [27] | Individualized Exercise Training [27] | Fioritto et al., 2021 |
|---|---|---|---|---|---|
| Lower Limb Muscle Strength | 1 min STS [27] | Feasibility [27] | Hospital [27] | Individualized Exercise Training [27] | Fioritto et al., 2021 |
| Physical Function [47] | The 6-Minute Walk Test [47] Timed Up and Go Test [47] The Physical Performance Test [47] | RCT [47] | Hospital [47] | Multidirectional Walking [47] | Potiaumpai et al., 2020 |

### 3.1. Core Area Feasibility

In the core area of "Feasibility", $n = 8$ different outcomes were measured 30 times *using n* = 15 different instruments (Table 2). The outcome feasibility was the most frequently measured outcome in this core are. It was measured two times in studies that only included allogeneic HSCT patients and seven times in studies including mixed HSCT patients.

### 3.2. Core Area Life Impact

In the core area "life impact", $n = 37$ different outcomes were measured 105 times *using n* = 49 different instruments (Table 2). Fatigue was the most frequently measured outcome *(n* = 15) in all of the studies, regardless of design, setting, or the included population. It was measured using $n= 12$ different instruments. In studies that only included allogeneic HSCT patients, fatigue was measured 8 times using $n = 7$ different instruments. Studies including mixed HSCT patients measured fatigue 7 times using $n = 8$ different instruments.

Quality of Life *(n* = 5) and Health Related Quality of Life *(n* = 4) were measured 9 times *using n* = 2 different instruments in studies that only included allogeneic HSCT patients. The most frequently used instrument used to measure quality of life in this population was the EORTC QLQ-C30 [55]. In studies including a mixed HSCT population, Quality of Life *(n* = 5) and Health related Quality of Life *(n* = 7) were measured 12 times *using n* = 6 different instruments.

Depression was measured 11 times in studies including allogeneic HSCT *(n* = 5) or mixed HSCT *(n* = 6) patients. The Hospital Anxiety and Depression Scale [56] was the most frequently used instrument used to measure depression in studies including mixed HSCT patients. Studies including allogeneic HSCT patients only *used n* = 6 different instruments.

Anxiety was measured eight times. In studies including allogeneic HSCT patients only, anxiety was measured *in n* = 3 studies *using n* = 3 different instruments. In studies including mixed HSCT patients, it was measured *in n* = 5 studies *using n* = 2 different instruments. The most frequently used instrument used to measure anxiety was the Hospital Anxiety and Depression Scale [56]. Anxiety was measured in seven out of eight studies during the "Hospital" phase.

### 3.3. Core Area Pathophysiology

In the core area "pathophysiological manifestations", 39 different outcomes were measured 85 times using 61 instruments (Table 2). Endurance *(n* = 4) and handgrip Strength *(n* = 4) were the most frequently used outcomes. Both outcomes were used two times in studies including both allogeneic HSCT patients only and mixed HSCT patients.

All four studies used a handgrip dynamometer to measure handgrip strength. Endurance was measured using five different instruments, always during the "Hospital" phase.

### 3.4. Timing of Measurement

In 23 out of 39 of the studies, measurements were performed at two time points (see Table 3). The maximum number of measurements was n = 7 time points. Regardless of the setting, the initial measurements (T1) were not always performed on admission. A total of 22 studies were conducted in a hospital setting; in n = 13 studies, measurements were performed on admission, while in n = 9 studies, measurements were not performed on admission. A total of 17 studies were conducted in a non-hospital setting; in n = 13 studies, measurements were performed on admission, and in n = 4 studies, measurements were not performed on admission.

**Table 3.** Timing of measurement.

| HOSPITAL SETTING | | | | | | | |
|---|---|---|---|---|---|---|---|
| Allogeneic HSCT | | | | | | | |
| | T1 | T2 | T3 | T4 | T5 | T6 | T7 |
| Before Hospitalization | [37] | | | | | | |
| On Admission | [16,17,30,31,34,36,37,52] | | | | | | |
| At Baseline | | | | | | | |
| At discharge | | [17,30,31,34,36,37] | | | | | |
| Before the intervention | [33] | | | | | | |
| After the intervention | | [33] | | | | | |
| First Session | [38] | | | | | | |
| Second Session | | [38] | | | | | |
| One week before HSCT | | [16] | | | | | |
| Day − 2 before HSCT | | | | | | | |
| Day − 1 before HSCT | | | | | | | |
| Day + 2 after HSCT | | | | | | | |
| Before HSCT | | | | | | | |
| After HSCT | | | | | | | |
| Second week of Hospitalization | | | | | | | |
| Day + 7 after HSCT | | | | | | | |
| Day + 8 after HSCT | | | | | | | |
| Day + 10 after HSCT | | | | | | | |
| Day + 14 after HSCT | | | | | | | |
| Day + 20 after HSCT | | | | | | | |
| Day + 30 after HSCT | | | | | | | |
| 7 weeks after Hospitalization | | | [37] | | | | |
| Day + 60 after HSCT | | | | | | | |
| 3 months after HSCT | | | [17] | | | | |
| Day + 100 after HSCT | | [52] | [16] | | | | |
| 6 months after HSCT | | | [30] | | | | |
| 9 months after HSCT | | | | | | | |
| One year after HSCT | | | | [16] | | | |
| Hospital Setting | | | | | | | |
| HSCT | | | | | | | |
| | T1 | T2 | T3 | T4 | T5 | T6 | T7 |

| | T1 | T2 | T3 | T4 | T5 | T6 | T7 |
|---|---|---|---|---|---|---|---|
| Before Hospitalization | [40] | | | | | | |
| On Admission | [20,23,34,39,44] | | | | | | |
| At Baseline | [27,45,47] | | | | | | |
| At discharge | | [18,20,27,34] | | [44] | | | |
| Before the intervention | [18,19] | | | | | | |
| After the intervention | | [19] | | | | | |
| First Session | | | | | | | |
| Second Session | | | | | | | |
| One week before HSCT | | [47] | | | | | |
| Day − 2 before HSCT | [46] | | | | | | |
| Day − 1 before HSCT | [54] | [44] | | | | | |
| Day + 2 after HSCT | | [54] | | | | | |
| Before HSCT | | | | | | | |
| After HSCT | | | | | | | |
| Second week of Hospitalization | | [45] | | | | | |
| Day + 7 after HSCT | | | [44,54] | | | | |
| Day + 8 after HSCT | | [39] | | | | | |
| Day + 10 after HSCT | | [46] | | | | | |
| Day + 14 after HSCT | | [23] | [39] | | | | |
| Day + 20 after HSCT | | | [46] | | | | |
| Day + 30 after HSCT | | | [23,47] | | | | |
| 7 weeks after Hospitalization | | [40] | | | | | |
| Day + 60 after HSCT | | | | | [44] | | |
| 3 months after HSCT | | | [45] | | | | |
| Day + 100 after HSCT | | | | [23] | | [44] | |
| 6 months after HSCT | | | | [45] | | | [44] |
| 9 months after HSCT | | | | | | | |
| One year after HSCT | | | | | | | |
| NON-Hospital Setting | | | | | | | |
| Allogeneic HSCT | | | | | | | |
| | T1 | T2 | T3 | T4 | T5 | T6 | T7 |
| Before Hospitalization | | | | | | | |
| On Admission | [28] | | | | | | |
| At Baseline | | | | | | | |
| At discharge | | [28] | | | | | |
| Before the intervention | [29,32,35] | | | | | | |
| After the intervention | | [29,32,35] | | | | | |
| First Session | | | | | | | |
| Second Session | | | | | | | |
| One week before HSCT | | | | | | | |
| Day − 2 before HSCT | | | | | | | |
| Day − 1 before HSCT | | | | | | | |
| Day + 2 after HSCT | | | | | | | |
| Before HSCT | | | | | | | |
| After HSCT | | | | | | | |
| Second week of Hospitalization | | | | | | | |
| Day + 7 after HSCT | | | | | | | |
| Day + 8 after HSCT | | | | | | | |
| Day + 10 after HSCT | | | | | | | |

| | T1 | T2 | T3 | T4 | T5 | T6 | T7 |
|---|---|---|---|---|---|---|---|
| Day + 14 after HSCT | | | | | | | |
| Day + 20 after HSCT | | | | | | | |
| Day + 30 after HSCT | | | | | | | |
| 7 weeks after Hospitalization | | | | | | | |
| Day + 60 after HSCT | | | | | | | |
| 3 months after HSCT | | | | | | | |
| Day + 100 after HSCT | | | | | | | |
| 6 months after HSCT | | | | | | | |
| 9 months after HSCT | | | | | | | |
| One year after HSCT | | | | | | | |
| Non-Hospital Setting | | | | | | | |
| HSCT | | | | | | | |
| Before Hospitalization | | | | | | | |
| On Admission | | | | | | | |
| At Baseline | [21,22,24,25,41,48–50] | | | | | | |
| At discharge | | [22,41,42] | | | | | |
| Before the intervention | [42,43] | | | | | | |
| After the intervention | | [21,24,43] | [41] | | | | |
| First Session | | | | | | | |
| Second Session | | | | | | | |
| One week before HSCT | | [25] | | | | | |
| Day − 2 before HSCT | | | | | | | |
| Day − 1 before HSCT | | | | | | | |
| Day + 2 after HSCT | | | | | | | |
| Before HSCT | [53] | | | | | | |
| After HSCT | | [53] | | | | | |
| Second week of Hospitalization | | | | | | | |
| Day + 7 after HSCT | | | | | | | |
| Day + 8 after HSCT | | | | | | | |
| Day + 10 after HSCT | | | | | | | |
| Day + 14 after HSCT | | | | | | | |
| Day + 20 after HSCT | | | | | | | |
| Day + 30 after HSCT | | [50] | | | | | |
| 7 weeks after Hospitalization | | | | | | | |
| Day + 60 after HSCT | | | [50] | | | | |
| 3 months after HSCT | | | | | | | |
| Day + 100 after HSCT | | | | [50] | | | |
| 6 months after HSCT | | [48,49] | | | [50] | | |
| 9 months after HSCT | | | [48,49] | | | | |
| One year after HSCT | | | | [48,49] | | | |

## 4. Discussion

In this review, we observed a tendency toward the use of the same specific outcomes and outcome measurement instruments within the two core areas Feasibility and Life Impact; however, we saw a much more diverse use of outcomes and tools in the core area "Pathophysiological Manifestations". Despite the use of the same outcomes and outcome measurement instruments, the scientific efforts in this field do not fully exploit the potential for evidence synthesis, clinical interpretation, and constructive implications for

further research. The main reasons for this are measurement bias due to the heterogeneity and inconsistency of outcomes and outcome measurement instruments used, which is in line with similar statements in the COMET Handbook [57] that describe problems related to outcome reporting bias and inconsistency in outcome measurement. Below, we discuss four main aspects of measurement bias that we encountered based on our results.

## 5. Outcome Excess and Inconsistent Use

The 84 different outcomes that were measured in the studies that we included in this scoping review as well as the wide variety of terms used for the same outcomes indicate an excess of outcomes and the inconsistent use of terms in the body of literature that we reviewed. For example, in the "Pathophysiology" core area, thirteen terms were used to describe similar outcomes, of which we only recognize three distinct outcomes, all of which are related to, in different degrees, the body's capacity to produce energy through aerobic metabolic pathways (peak aerobic capacity, peak oxygen consumption, aerobic fitness, functional aerobic capacity, cardiorespiratory fitness, and aerobic endurance performance capacity) or to move itself in a specific manner within a specific timeframe (exercise capacity, functional exercise capacity, maximal exercise capacity, submaximal exercise capacity, and endurance) as well as a third more complex outcome that includes multiple components of fitness (physical capacity and physical performance).

This heterogeneous use of terminology hampers communication between researchers and impedes synthesis in secondary research. It also generates confusion concerning the content of each outcome, which could lead to aberrant inclusions or exclusions in reviews or even incorrect interpretations by clinicians.

Researchers in the field of rehabilitation for patients treated with allogeneic HSCT should seek to reduce the number of the outcomes they measure by reaching consensus about the relevant outcomes to be collected and reported, thus defining a core outcome set (COS). Ideally, COS development should involve patients, so that their needs and insights are taken in consideration.

Strength is an important outcome in the core area "Pathophysiology" because its reduction due to corticosteroid regimens can determine functional performance in post-allogeneic HSCT long-term survivors [58,59]. Handgrip strength can be used as a surrogate marker of strength among patients undergoing allogeneic HSCT, and it can detect strength loss and be regained post-allogeneic HSCT [60]. It is a widely used outcome in HSCT research, something that is probably due to the practicability of its measurement. Other authors underline the importance of this outcome during hospitalization for allogeneic HSCT since detecting strength loss can improve fall prevention [61]. However, in addition to handgrip strength, eight other aspects of strength were measured in the studies that we included (i.e., isokinetic leg performance, knee extension strength, muscle strength, peripheral muscle strength, strength, strength capacity, trunk strength, and upper limb muscle strength). As a result, again, there is heterogeneity in the outcomes being measured, which hampers synthesis and adds data waste to this research field. Given the importance of the outcome strength for patients treated with HSCT, researchers should reach consensus on which aspect of strength is the most relevant to be measured.

In the "Feasibility" core area, we observed the interchangeable use of terms (for example, "accrual acceptance", "acceptability", "rate of participant enrolment", "recruitment", and "recruitment rate") since similar terms were used to describe identical phenomena. The most frequently used outcome in this core area—the outcome feasibility—is in our view, a multidimensional construct that comprises dimensions such as safety, attrition, acceptability, and adherence. Some researchers in the field of allogeneic HSCT rehabilitation have already begun to approach feasibility in the manner in which we see it [14,62]. In this review, we noticed that various authors classified specific terms as distinct outcomes (i.e., "acceptability", "adherence", and "attrition"), while others used these terms as instruments to measure the outcome feasibility. This difference in

definitions and outcome operationalization leads to incomparable data and is a waste of resources.

Dimensions such as safety, attrition, acceptability, and adherence should not be considered outcome measurement instruments and should not be used and reported as such because they refer to what is measured, i.e., an outcome, while an instrument refers to how an outcome is measured. Ideally, the research community in this field should reach a consensus on the definition of feasibility and on how to measure it.

## 6. Outcome Measurement Instrument Excess and Inconsistent Use

The 84 outcomes that were found were measured by 134 different measurement instruments. In the "Pathophysiology" core area alone, 59 different instruments were used to measure 39 different outcomes. This diverseness in the outcome measurement instruments indicates an excess of outcome measurement instruments.

This excess of outcome measurement instruments makes synthesis across studies more difficult. A meta-analytical systematic review studying the effects of physical activity on fatigue confirms our statement [63]. In that study, the authors had to describe intervention effects using standardized mean differences— which are more difficult to interpret—rather than weighted mean differences, because the studies that they reviewed used different outcome instruments to measure fatigue.

Patients undergoing allogeneic HSCT commonly experience fatigue both during hospitalization and in the long-term [64]. Different items could be relevant to measure fatigue in one situation but not in the other since fatigue during hospitalization (i.e., cancer treatment related fatigue) may have different characteristics than long-term fatigue (i.e., cancer-related fatigue). However, the variety of instruments used to measure fatigue remains wide, making comparing fatigue measurements difficult. An item response theory (IRT) -based item bank, such as the Patient-Reported Outcomes Measurement Information System (PROMIS) [65] Fatigue Item bank, could address problems related to measuring different levels of fatigue, as tailored shortforms for different patient populations can be developed or computer adaptive testing could be used.

The variety of outcome instruments has a positive impact when it serves the practicability of measurement conduction in different settings and phases. For example, in our review, we found that ($n$ = 5) different instruments were used to measure the outcome "endurance." Patients treated with allogeneic HSCT are unable to perform the six-minute walk test or the cardiopulmonary exercise testing during hospitalization, as they are generally restricted to their rooms to reduce the risk of infection and because they are connected to medication-administering devices. In this case, an endurance test that can be performed in a small space, such as the six-minute step test, has better practicability than the six-minute walk test. The appropriate use of a wide variety of outcome measurement instruments requires specific context- and phase-including guidelines, which would serve the avoidance of inconsistent scientific output. Ideally, such guidelines should be informed based on clinimetric studies to confirm the reliability and validity of the indicated instruments in defined settings and phases.

We noticed that some instruments such as the EORTC QlQ-C30 and the FACT were often used to measure distinct outcomes such as Health-Related Quality of Life and Quality of Life [66]. We made the same observation for the six-minute walk test, which was used to measure different outcomes. Using a single measurement instrument to measure different outcomes is often not a correct practice because the measurement properties of a measurement instrument may be sufficient to measure one outcome but insufficient to measure another outcome. Therefore, before use, researchers should ensure that the clinimetric properties of each outcome measurement instrument are appropriate for measurement in the population of interest.

## 7. Timing and Setting of Measurement Inconsistency

In this review, we found notable heterogeneity in the timing of measurements across studies. Our findings confirm those of van Haren et al. [67] that time-point heterogeneity does not allow for follow-up measurement synthesis in systematic reviews. The general condition of patients treated with allogeneic HSCT fluctuates depending on the phase of their treatment. At the beginning of hospitalization, they may be sturdy, but, later on and depending on chemotherapy intensity, they may suffer from severe fatigue, infection symptoms, and nutritional deficits due to mucositis or other reasons. When patients begin to recover, they gradually show an improved general condition. However, those who suffer from severe symptoms during hospitalization are usually weaker at discharge than at admission. Therefore, heterogeneity in the timing of measurements is an important source of bias since timing is associated with the general condition of the patient. For example, if the "baseline" measurements of one study are performed on admission and the final measurements are performed at discharge, then the results of these measurements or their differences are incomparable to those of another study in which the measurements were performed at day four or ten after admission and at three months after discharge.

Due to the fluctuating condition of patients treated with allogeneic HSCT, not all measurements are always feasible or even meaningful across settings. Measurements might have less value for patients, increase their workload during a period in which filling in questionnaires is not their highest priority, add to data waste, and increase heterogeneity in measurement timing. In order to avoid unnecessary patient effort and the production of data waste and in an effort to improve our understanding of phenomena with established clinical significance, researchers should agree on some basic assumptions: (a) the phases they recognize in the process of allogeneic HSCT (i.e., before allogeneic HSCT, during hospitalization, 100 days after allogeneic HSCT, one year after HSCT—Van der Lans et al. have already made efforts to recognize different phases based on patient insights during recovery) [68]; (b) the outcomes to be measured in each phase; and (c) the timing at which the measurements for each outcome are taken and the method used to measure them in each phase.

## 8. Allogeneic HSCT vs. HSCT Population

In this review, we found that 64% of the reported research projects recruited both allogeneic HSCT and autologous HSCT patients. There are some arguments for combining these populations in a study, though there are no formal restrictions at all since the EBMT Handbook [69] does not even have a dedicated article on rehabilitation from which arguments for the distinction of these two populations could arise. Both populations suffer from haematological malignancies, and both populations undergo transplantation. Therefore, researchers in the field of rehabilitation include samples from both populations to achieve the targeted sample size much more quickly.

However, major differences exist between these two populations, which could lead to problems during the interpretation of study results. First, although both undergo "transplantation", the two populations do not undergo the same medical treatment. Chemotherapeutic and, more importantly, immunosuppressive treatments differ with regard to duration and side effects. Second, allogeneic HSCT patients normally undergo a longer and more strict isolation period in addition to a longer planned hospital stay. Third, allogeneic HSCT patients often suffer from GvHD and require additional medical treatment, resulting in significant physical and psychological deterioration.

Consequently, these two different populations cannot be combined in research due to differences in measurement timing and the relevance of the outcomes.

There are many published studies indicating that patients from both populations have been recruited. However, the scientific community should consider whether

recruiting patients from both populations is appropriate practice and should reach consensus concerning future practice.

## 9. Limitations

To our knowledge, this review is the first attempt to describe the outcomes and measurement instruments used in the study of rehabilitative interventions for patients undergoing allogeneic HSCT. Although we managed to elucidate major issues concerning heterogeneity in the outcomes and measurement instruments used, our findings must be interpreted in light of the limitations of this review. First, we only included interventional studies and we only included research published in German and English. This strategy may have prevented the retrieval and inclusion of publications in other languages and from a wider range of disciplines. As a result, this scoping review focuses on the main body of work on psychological and physical rehabilitative interventions. Second, we classified the outcomes we retrieved based on two different frameworks, as the Boers et al. framework was designed for another purpose and thus does not offer a distinct classification for feasibility outcomes. Finally, we extracted and classified outcomes and instruments according to the terms used by the authors, without modification or interpretation, and therefore, the extracted terms were not always appropriate.

## 10. Conclusions

Research in the field of rehabilitation for patients with haematological malignancies treated with allogeneic HSCT covers measurements in all relevant core areas. However, this field of study is hampered by excess outcomes and inconsistent outcome terminology. Furthermore, we detected the inconsistent use of measurement instruments in terms of setting and timing. The combined recruitment of allogeneic and autologous HSCT patients may exacerbate these problems, thus reducing the successful exploitation of the study results by hampering synthesis and clinical interpretation. We recommend that researchers reach a consensus with regard to the use of common terminology for the outcomes of interest and homogeneity in measurement instrument selection and measurement timing.

**Author Contributions:** A.I.M.: conceptualization, data curation, formal analysis, investigation, methodology, project administration, validation, writing—original draft; P.T.: conceptualization, data curation, formal analysis, investigation, methodology, project administration, validation, writing—original draft; D.K.: data curation, formal analysis, investigation, validation; L.B.M.: conceptualization, data curation, formal analysis, investigation, methodology, project administration, validation, writing—original draft. All authors have read and agreed to the published version of the manuscript.

**Funding:** This research received no external funding.

**Conflicts of Interest:** The authors declare no conflict of interest.

## Abbreviations

| | |
|---|---|
| HSCT | Hematopoietic Stem Cell Transplantation |
| GvHD | Graft Versus Host Disease |
| IST | Immunosuppressive Therapy |
| PRISMA | Preferred Reporting Items for Systematic Reviews and Meta-Analyses |
| EORTC | European Organization for Research and Treatment of Cancer |
| COS | Core Outcome Set |
| IRT | Item Response Theory |
| PROMIS | Patient-Reported Outcomes Measurement Information System |
| FACT | Functional Assessment of Cancer Therapy |
| EBMT | European Society for Blood and Marrow Transplantation |

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
