# Peer review of "A Scoping Review on Outcomes and Outcome Measurement Instruments in Rehabilitative Interventions for Patients with Haematological Malignancies Treated with Allogeneic Stem Cell Transplantation"

_curroncol, doi:10.3390/curroncol29070397_

Round 1
Reviewer 1 Report
This report adresses an issue that is poorly standardized, i.e., the rehabilitation of allogeneic SCT patients. As the Authors appropriately state, research in the field of rehabilitation for patients with hematological malignancies treated with allogeneic (HSCT) is hampered by the excess out-comes used, the inconsistent outcome terminology, and the inconsistent use of measurement instruments in terms of setting and timing.
The Review is well organized and documented. I have two observations: Table 2, which is very detailed, is difficult to read. Splitting the Table according to Core Areas may be useful.The second observation is that, in the Discussion, is worth mentioning that the EBMT Handbook (which is a widely utilized handbook on HSCT) does not include a chapter on rehabilitation.
Author Response
Dear sir or madam,
Thank you for your review!
As far as your comments are concerned:
a) You are right! One should split the tables according to core areas. We actually did split the tables according to core areas AND population. The reason you don't see it has to do with the layout standards of the journal. We could not even use bold for the headings to help the reader understand when we change core area or population. However, I believe that the journal specialists have good reasons to adhere on this layout.
b) Thank you for this comment concerning the EBMT Handbook. I will add it in the manuscript, and I will cite the EBMT Handbook.
Once again thank you for your review and best regards!
Reviewer 2 Report
Allogeneic Stem cell transplantation still represents the only curative option in many hematologic malignancies, despite the significant increase of conventional drugs. Over the past 15 years the improvement of HLA testing, the increase of available donors, the possibility of an intensity modulation of the preparative regimens has extended the procedure also to older age, until now ineligible, populations. Although many score systems has been developed to predict transplant toxicity and survival probability, according the improvement of patients care, very less has been made to evaluate post transplantation “quality of life”.
This paper focused on the outcome measurement tools after allogeneic transplantation for hematological malignancies, reviewing 39 studies and analyzing different outcome variables and different measurement instruments.
Despite some limitations, the paper is very interesting and provides an updated picture of the disappointing “state of art” of post-transplant rehabilitation.
The authors underlines the huge variability in terminology, in outcome variables, in rehabilitative start time, and pave the way for studies aiming the homogenization of measurement tools/interventions.
Author Response
Dear sir or madam,
Thank you for your review and for your comments. Your comments seem to confirm my views as expressed in the rationale and they recognize the value of this work for the scientific society. I could not reveal any specific suggestions for revisions in your comments. Thank you very much for your review and for the recognition of the hard and detailed work behind this paper.
Best regards!